# Yield, Quality and Physiological Traits of Red Beet Under Different Magnesium Nutrition and Light Intensity Levels

**Sara D'Egidio [1], Angelica Galieni [2], Fabio Stagnari [1,*], Giancarlo Pagnani [1] and Michele Pisante [1]** 

[1] Faculty of Bioscience and Technologies for Food, Agriculture and Environment, University of Teramo, Campus Universitario di Coste Sant'Agostino, via R. Balzarini 1-64100 Teramo (TE), Italy

[2] Council for Agricultural Research and Economics - Research Centre for Vegetable and Ornamental Crops, Via Salaria 1-63077 Monsampolo del Tronto (AP), Italy

\* Correspondence: fstagnari@unite.it; Tel.: +39-0861-266914; Fax: +39-0861-266914

**Abstract:** The effects of light intensity and Magnesium (Mg) supply on quality traits, yield and macronutrient assimilation of red beet plants were studied in two greenhouse experiments (in 2017 and 2018). According to a split-plot design, we compared two photosynthetically active radiation (PAR) levels (100% PAR, Full Light, FL and 50% PAR, Light Reduction, LR) as the main factor and three Mg application rates (0, 30, and 60 kg Mg ha$^{-1}$: MG_0, MG_30 and MG_60, respectively) as the secondary factor. Yield and dry matter accumulations were principally affected by Mg. In both growing seasons, storage root dry weight (DW) increased about 5-fold in MG_60 with respect to MG_0; the highest leaves DW was achieved with the "LR × MG_60" treatment. Nitrogen and Mg contents in leaves and storage roots increased as Mg availability increased; also, the highest chlorophyll content was obtained combining LR and a high Mg rate. Moreover, the reflectance-derivative Normalized Difference Vegetation Index (NDVI$_{670}$) and Chlorophyll Index (CI) allowed for discriminating the Mg sub-optimal supply in red beet plants. Sucrose was found to be the most abundant sugar in both the leaves and storage organs and was affected by Mg supply. Total phenolic content and betalains in storage roots at harvest were affected by both PAR and Mg application rates. Our results highlight the potential of Mg nutrition in ensuring good yield and quality of red beet crops.

**Keywords:** Mg sub-optimal availability; PAR × Mg interaction; red beet

## 1. Introduction

The chemical composition of vegetables depends on a number of factors, including plant genotype and growing conditions; the manipulation of crop management could represent a way to sharply induce the modification of edible organs, obtaining an improved quality of the final products [1–4]. In particular, fertilization plays a key role since the accumulation of plant metabolites is closely associated with the availability of mineral elements in the growing substrates [5,6]. Although the impact of organic and basic mineral (nitrogen, N, phosphorus, P, and potassium, K) fertilization on yield and quality traits of vegetables is thoroughly investigated, little is known about the effect of magnesium (Mg). Mg is an important macronutrient for plant nutrition [7]. It is the second most abundant cation in cells and is involved in a wide range of biological activities, operating as the activator or regulator of many key enzymes in plant physiological processes [8]. As a central element of the chlorophyll (Chl) structure, Mg is important in the harvest of solar energy; it also plays a critical role in phloem loading and the transportation of photoassimilates into sink organs, such as fruits, roots and seeds [9]. Mg deficiency results in (i) suppressing plant growth [8], as it directly affects photosynthesis [10,11]; (ii)

reducing dry matter partitioning between roots and shoots; (iii) enhancing sugar, starch and amino acid accumulation in leaves; (iv) breakdown of Chl molecules; (v) over-reduction in the photosynthetic electron transport chain and generation of highly reactive oxygen species (ROS) due to the impairment in photosynthetic $CO_2$ fixation [9,12]. It emerges that adequate Mg nutrition is required for the maintenance of high-quality yields.

The responses of plants to different Mg amounts are affected by environmental conditions, i.e. light intensity, temperature regimes and genotypes [13]. In particular, Mg deficiency symptoms are highly dependent on light availability which, at high intensities, increases the development of interveinal chlorosis and species-specific symptoms, such as some reddish spots on the leaf blade (characteristic of red beet plants). Besides, light is in any case an important factor, affecting both photosynthetic light- and carbon-use efficiency, which in turn influences plant yield and quality [14,15]: shading reduces dry matter, favors biomass allocation to the aerial parts and alters photosynthetic pigments by reducing the Chl A: Chl B ratio [16–20]. It follows that studies on the interaction between light availability and mineral nutrition to optimize crop productivity and quality and to avoid excessive application of fertilizers are required.

Leaf spectral reflectance could be very useful for early, easy and inexpensive estimation of sub-optimal growing conditions, such as those observed under mineral deficiencies. Nutrient deficiencies are indeed responsible for decreased leaf chlorophyll concentration, thus modifying the reflectance in the visible and near-infrared range and shifting the red-edge position to shorter wavelengths. Several vegetation indices have been proposed to estimate changes in Chl content in leaves. Besides the well-known Normalized Difference Vegetation Index (NDVI), other reflectance-based indices such as the Photochemical Reflectance Index (PRI) - an important spectral index for the monitoring of efficiency of photosynthetic radiation use in plants [21] - and the Chlorophyll Index (CI) - which successfully estimates high Chl concentration levels in leaves [22–24] - are widely used.

With this work, we studied the effects of Mg supply under different photosynthetically active radiation (PAR) availabilities. We selected red beet (*Beta vulgaris* var. conditiva Alef.) as a model species possessing a sucrose storage sink (see also Hermans et al. [12]) as well as it is grown for different food uses (pickles, salad, juice). Red beet is characterized by pro-health properties thanks to its phytonutrients content, such as minerals, sugars, dietary fiber, vitamins, fatty acids, polyphenols and betalains. In particular, betalains-water-soluble nitrogen-containing pigments, particularly appreciated as natural food colorants in view of their potential as antioxidant pigments, which represents a group of phytochemicals that is scarcely introduced in the diet. Only a few plant species can supply betalains, and their use as a food colorant or to enrich functional foods could increase their consumption [25]. As a consequence, raw materials characterized by higher betalains content could be appreciated for both industry and fresh consumption (although studies on bioavailability and metabolism of betalains are scarce [25]).

The objective of the research study was to investigate the response of red beet plants to the combinations PAR availability/Mg supply in terms of yield, quality and physiological traits. The biochemical characterization of storage organs was performed to obtain useful knowledge to define the optimal Mg/PAR availability combination, which allows obtaining high-quality products, characterized by elevated macronutrients concentration (especially Mg) to improve dietary intake.

## 2. Materials and Methods

### 2.1. Plant Material, Experimental Design and Growing Condition

Two experiments were conducted at the Department of Biosciences and Technologies for Agriculture, Food and Environment, University of Teramo (Italy) from 01-02-2017 to 25-05-2017 and from 15-02-2018 to 08-06-2018, under greenhouse conditions. The greenhouse was covered with a single layer of ethylene-vinyl acetate (PATILUX) provided by P.A.T.I. S.p.A. (San Zenone degli Ezzelini, TV, Italy); no artificial lights, fans and heaters were provided. In both years, the environmental

conditions were monitored starting from transplanting and all along the crop cycle with radiation, air temperature and humidity sensors, all connected to a data logger (EM50 Data Collection System, Decagon Devices Inc., Pullman, WA, USA).

Seeds of red beet (*Beta vulgaris* L. var. conditiva Alef., *cv.* "Tonda di Chioggia", L'Ortolano S.r.l., Cesena, FC, Italy) were sown on a nursery potting soil (Huminsubstrat N3, Neuhaus, Klasmann-Deilmann, Geeste, Germany) composed of 90% peat and 10% clay. When the first two true leaves were fully expanded [40 and 48 days after sowing (DAS) in 2017 and 2018, respectively], the seedlings were transplanted into 16 cm diameter plastic pots (3 L) filled with a mixture of coconut coir, sand and perlite (3:1:1, v/v) at a density of 1 plant per pot.

The experimental design was a split-plot with three replications (30 pots per replication), as follows: two levels of PAR ($\mu$mol m$^{-2}$ s$^{-1}$) availability (main plots) and three Mg application rates (subplots); six experimental treatments, obtained by the combination of the two experimental factors, were compared. The two levels of PAR availability consisted of (i) a Full Light (unshaded) treatment (100% of PAR availability, FL); (ii) a shaded treatment (50% of PAR availability, Light Reduction, LR). PAR was 781 and 405 $\mu$mol m$^{-2}$ s$^{-1}$ in FL and LR, respectively, averaging over both years and growing cycles. Shading was accomplished using a black shade net (provided by CARRETTA tessitura s.n.c., Carrè, VI, Italy) which was wrapped around a rigid and removable structure placed above the vegetation (till to 5 cm above the bottom of the pots) in order to intercept the incoming light from the top and sides. To allow air circulation, light was not limited from below. Shading nets were applied starting from 5 days after transplanting (DAT). PAR intensity was measured every 15 min and the % of shading was determined by comparing the average PAR values of net with the average PAR values of the un-shaded treatments (natural sunlight under greenhouse). Under the glasshouse, the % of reduction with respect to outdoors conditions in terms of total solar radiation (W m$^{-2}$) and PAR ($\mu$mol m$^{-2}$ s$^{-1}$) amounted to 7.4% and 12.6%, respectively. The mean air temperatures recorded during the entire crop cycle were 16.3 °C and 15.8 °C in FL and LR, respectively, in 2017 and 19.9 °C and 19.1 °C in FL and LR, respectively, in 2018. Tiny differences were observed among shading conditions and also for relative humidity (RH), with averaged values of 65.6% and 74.0% in 2017 and 2018, respectively.

Mg application rates consisted of: (i) no Mg application (0 kg ha$^{-1}$, MG_0); (ii) 30 kg Mg ha$^{-1}$ (MG_30) and (iii) 60 kg Mg ha$^{-1}$ (MG_60). The highest Mg rate was used as it represents the most commonly applied rate for sugar beet in many European countries [26]. The established Mg levels were obtained by applying 0.24 g and 0.48 g pot$^{-1}$ of MagnesiumFast (©Cifo S.r.l, S. Giorgio di Piano, BO, Italy) - composed by 6% water-soluble magnesium oxide (MgO) and 12% water-soluble sulphur anhydride (SO$_3$) at 5 (1/3 of the dosage) and 12 (2/3 of the dosage) DAT. In order to standardize the amount of sulphur applied to the plants, MG_0 and MG_30 treatments were respectively fertilized with 137.5 and 68.75 kg S ha$^{-1}$, using the commercial product UMIZOL (Mormino S.r.l., Termini Imerese, PA, Italy).

### 2.2. Crop Management

At transplanting, the growth substrate was saturated with tap water before planting operation and each pot was supplemented with the commercial nutrient solution Floral 20-20-20 (©Cifo S.r.l., S. Giorgio di Piano, BO, Italy) at the dose of 15 kg ha$^{-1}$; the nutrient solution was composed as follows: 20% total N; 20% ammonium citrate; 20% water soluble phosphoric anhydride (P$_2$O$_5$); 20% potassium oxide (K$_2$O); 0.05% water-soluble boron (B); 0.01% water-soluble copper (Cu); 0.01% Cu chelated by EDTA; 0.20% water-soluble iron (Fe); 0.20% water-soluble Fe chelated by DTPA; 0.10% water-soluble manganese (Mn); 0.10% Mn chelated by EDTA; 0.005% water-soluble molybdenum (Mo); 0.01% water-soluble zinc (Zn); 0.01% Zn chelated by EDTA.

The pots were manually re-watered with tap water. No insecticide or fungicide treatments were performed.

## 2.3. Aerial and Belowground Biomass Accumulation

Three plants per replication were harvested at maturity (714 and 724 growing degree days (GDD, °C) in 2017 and 2018). GDD were calculated as the accumulation of mean air temperature exceeding the base temperature of 5.6 °C [27]. The sampled plants were separated into leaves and storage roots, weighed for fresh weights (FW) determinations and freeze-dried to obtain the organ's dry weights (DW, g plant$^{-1}$).

## 2.4. Leaf Reflectance and Soil-Plant Analysis Development (SPAD) Readings

The reflectance was measured with a HandHeld 2 Pro Portable FieldSpec Spectroradiometer (ADS Inc., Boulder, CO, USA) from the leaves of 9 plants per treatment at 143, 358 and 714 GDD in 2017 and at 157, 376 and 724 GDD in 2018. The instrument measures spectra over a spectral range of 325–1075 nm, covering the visible and part of the near infrared portion of the spectrum, with an accuracy of ±1 nm. To minimize the effects of the sun's position, the reflectance measurements were taken within 1 h, near solar noon. Starting from the reflectance data, the following indices, which estimate leaf Chl content, were calculated:

NDVI$_{670}$ [28]:

$$NDVI_{670} = \frac{(R800 - R670)}{(R800 + R670)}$$

PRI [29,30]:

$$PRI = \frac{(R531 - R570)}{(R531 + R570)}$$

CI [23,24]:

$$CI = \frac{(R750 - R705)}{(R750 + R705)}$$

At the same GDD, Chl estimation was also performed with SPAD, using a 502 plus portable chlorophyll meter (Konica Minolta, Inc., Tokyo, Japan) in the mid-section of 3 fully expanded and same sun-oriented leaves per experimental unit.

## 2.5. Analytical Determinations

Sampling materials for analytical determinations were washed with deionized water, cut into small pieces and stored at −20 °C until freeze-dried. The freeze-dried material was then reduced into powder with a mortar and pestle using liquid nitrogen and was used for comparative analysis.

### 2.5.1. Total Nitrogen and Magnesium Determination, Chlorophyll Content

At 714 and 724 GDD in 2017 and 2018, respectively, sub-samples of both leaves and storage roots were characterized for their N and Mg concentrations.

N concentration was estimated with the Kjeldahl method; analysis was performed in triplicate and results were expressed as mg g$^{-1}$ DW.

An Agilent 7800 Inductively Coupled Plasma (ICP) mass spectrometer was used (Agilent Technologies, Waldbronn, Germany) for Mg determination (expressed as mg g$^{-1}$ DW). Samples were digested in closed vessels in an Automatic Microwave Digestion System (model MARS-5, CEM Corporation, Matthews, NC, USA). Then, 0.05 g of leaves and 0.15 g of storage roots were accurately weighed and transferred to a Teflon container; 4 mL of $HNO_3$, 2 mL of 30% $H_2O_2$ and 2 mL of high purity deionized water were added. After the microwave digestion cycle, a solution containing 3% $HNO_3$ and 0.5% HCl was added to adjust the final volume to 50 mL; before analysis, all samples were diluted with the same solution.

At 143, 358 and 714 GDD and 157, 376 and 724 GDD in 2017 and 2018, respectively, the chlorophyll a (ChlA) and chlorophyll b (ChlB) contents in fully-expanded leaves (or the more expanded leaf, early in the crop cycle) were determined according to Arnon et al. [31]. For each experimental unit, two

cyclic leaflets (10 mm diameter) were obtained from fresh leaf and Chl was extracted using an 80% acetone solution. The extracts were then read at 663 nm and 645 nm for ChlA and ChlB, respectively, using a spectrophotometer (Jenway 6300, Jenway, Stone, UK). The pigments' contents, expressed as µg mg$^{-1}$ FW, were then calculated as follows:

$$ChlA = [(12.7) \times (Abs_{663})] - [(2.69) \times (Abs_{645})]$$

$$ChlB = [(22.9) \times (Abs_{645})] - [(4.68) \times (Abs_{663})]$$

### 2.5.2. Individual Carbohydrates

At 714 and 724 GDD, leaves and storage roots were analyzed for carbohydrates content (sucrose, glucose and fructose, expressed as g 100 g$^{-1}$ DW) according to Karkacier et al. [32] with some modifications: 0.3 g leaves and 0.5 g storage roots were dissolved in 8 mL and 10 mL, respectively, of distilled water and sonicated for 30 min at room temperature. After vortexing, the mixture was centrifuged at 4000 rpm for 5 min, and the supernatants were filtered through a PTFE (Polytetrafluoroethylene)0.45 µm membran filter transferred to a vial and analyzed using a Waters HPLC (Waters, Milford, MA, US).

For each analysis, the injection volume for each sample was 10 µL.

For the leaves, analysis was carried out using an Aminex HPX-87C column (300 x 7.8 mm, BioRad, Hercules, CA) and high purity deionized water as the mobile phase. The column temperature was maintained at 65 °C and the flow rate at 0.6 mL min$^{-1}$.

For the storage root, a Shodex Asahipak NH2P-50 column (Shodex, Tokyo, Japan) and a mobile phase composed of high purity deionized water and acetonitrile (75 / 25, v/v) were used. The column temperature was set at 30 °C and the flow rate at 0.6 mL min$^{-1}$.

### 2.5.3. Total Polyphenols Content (TPC) and Betalains Assay

Analysis were performed on storage roots sampled at 714 and 724 GDD in 2017 and 2018, respectively, in order to avoid any influences of mechanical stress (due, for example, to potential storage roots breakage). The amount of total polyphenols in the extracts was determined according to a modification of the Folin-Ciocalteu colorimetric method [33]. Specifically, 0.1 g of samples was dissolved in 1 mL of 80% methanol, sonicated with Sonis 4 for 40 min and then centrifuged at 4000 rpm for 10 min. The supernatant was collected and the same procedure was repeated once with a sonication time of 20 min. The combined supernatants were filtered through 0.45 µm PTFE syringe filters (Phenomenex, Torrance, CA, USA). An extract aliquot of 0.1 mL was introduced into a test tube and mixed with 0.5 mL of Folin-Ciocalteu reagent. After 3 min, 1.5 mL of 20% of sodium carbonate was added, made up to a volume of 10 mL, and the contents were mixed thoroughly. The solutions were incubated at room temperature for 2 h and centrifuged for 5 min at 4000 rpm; the absorbance of the extracts was measured at 765 nm (Jenway 6300, Jenway, Stone, UK). TPC was expressed as gallic acid equivalents (GAE) in milligrams per 100 g of dry material (mg GAE 100 g$^{-1}$ DW).

For the extraction of betalains, 0.1 g of the sample were dissolved in 1 mL of 50% ethanol, agitated for 20 min and finally centrifuged at 4000 rpm for 10 min. The supernatant was collected and the same was repeated one more time to ensure maximum betalains extraction.

The content of betacyanins and betaxanthins in the extracts was determined spectrophotometrically at 535 nm and 476 nm, respectively (Jenway 6300, Jenway, Stone, UK), and quantified according to the methods of Cardoso-Ugarte et al. [34].

### 2.6. Statistical Analysis

A two-way analysis of variance (ANOVA) was applied to test (F-test) the effects of the treatments. When significant differences were detected, the means were separated with Fisher's least significant

difference (LSD) test. In addition, standard error of the difference (s.e.d.) between means and standard errors of the means are shown in Tables 1–6 and Figure 1, respectively.

ANOVA assumptions were tested through graphical methods. The statistical analyses were performed using R software [31].

## 3. Results

### 3.1. Storage Root and Leaves Yields

A clear trend showed an increase in storage root yield as the Mg supply increased, regardless of PAR availability (Table 1). "LR × MG_60" and "LR × MG_30" produced the significantly highest maximum observed storage root yield values (7.84 g and 7.61 g plant$^{-1}$ in 2017 and 2018, respectively; Table 1); in 2018, "LR × MG_60" did not significantly differ to "LR × MG_30".

Leaves' DW at harvesting reflected the yield data. "LR × MG_60" produced the highest responses (1.63 g and 2.13 g plant$^{-1}$ in 2017 and 2018, respectively). PAR availability significantly influenced leaves' DW, mainly in 2018; however, although not always significant, LR induced higher leaves DW than FL, also in 2017 (Table 1).

**Table 1.** Leaves and storage root dry mass accumulation of red beet plants subjected to two levels of photosynthetically active radiation (PAR) availability (100% PAR availability: Full Light, FL; 50% PAR availability: Light reduction, LR) and three magnesium (Mg) application rates (0 kg ha$^{-1}$ of Mg: MG_0; 30 kg ha$^{-1}$ of Mg: MG_30; 60 kg ha$^{-1}$ of Mg: MG_60) recorded at maturity in 2017 and 2018 growing seasons. Measurements are the average values of three replications. Means followed by different letters (upper case letters: main effects; lower case letters: effects of interaction) significantly differ (Fisher's least significant difference (LSD), $p \leq 0.05$).

| | 2017 | | | | | | 2018 | | | | | |
|---|---|---|---|---|---|---|---|---|---|---|---|---|
| **Treatments** | **Leaves (g plant$^{-1}$)** | | | **Storage Root (g plant$^{-1}$)** | | | **Leaves (g plant$^{-1}$)** | | | **Storage Root (g plant$^{-1}$)** | | |
| | **FL** | **LR** | *o.m.*[†] | **FL** | **LR** | *o.m.* | **FL** | **LR** | *o.m.* | **FL** | **LR** | *o.m.* |
| *Leaves* | | | | | | | | | | | | |
| MG_0 | 0.68 | 0.72 | *0.70* C | 1.56 | 1.60 | *1.58* C | 0.88 | 0.86 | *0.87* B | 1.31 c | 1.17 c | *1.24* |
| MG_30 | 0.95 | 1.18 | *1.07* B | 6.05 | 6.15 | *6.09* B | 1.57 | 1.83 | *1.70* A | 4.90 b | 7.61 a | *6.25* |
| MG_60 | 1.52 | 1.63 | *1.58* A | 7.67 | 7.84 | *7.75* A | 1.57 | 2.13 | *1.84* A | 6.86 a | 7.42 a | *7.13* |
| *o.m.* | *1.05* | *1.18* | | *5.09* | *5.19* | | *1.34* B | *1.60* A | | *4.35* | *5.40* | |
| *PAR availability* | | *n.s.* | | | *n.s.* | | | ***(0.050)* | | | **(0.266)* | |
| *Mg rates* | | ***(0.034)* | | | ***(0.111)* | | | ***(0.146)* | | | ***(0.267)* | |
| *PAR × Mg* | | *n.s.* | | | *n.s.* | | | *n.s.* | | | *** (0.378)* | |

*o.m.*[†]: overall means. * $p < 0.05$; ** $p < 0.01$; *n.s.* = not-significant. In brackets: standard error of differences between means (s.e.d.). Degrees of freedom: PAR availability, 1; Mg rates, 2; PAR availability × Mg rates, 2; Residual, 8.

**Table 2.** N concentration (mg g$^{-1}$ DW) and Mg concentration (mg g$^{-1}$ DW) in leaves and storage roots of red beet plants subjected to two levels of photosynthetically active radiation (PAR) availability (100% PAR availability: Full Light, FL; 50% PAR availability: Light Reduction, LR) and three magnesium (Mg) application rates (0 kg ha$^{-1}$ of Mg: MG_0; 30 kg ha$^{-1}$ of Mg: MG_30; 60 kg ha$^{-1}$ of Mg: MG_60) at maturity in 2017 and 2018 growing seasons. Means followed by different letters (upper case letters: main effects; lower case letters: effects of interaction) significantly differ (Fisher's LSD, $p \leq 0.05$).

| Treatments | 2017 N (mg g$^{-1}$ DW) | | | 2017 Mg (mg g$^{-1}$ DW) | | | 2018 N (mg g$^{-1}$ DW) | | | 2018 Mg (mg g$^{-1}$ DW) | | |
|---|---|---|---|---|---|---|---|---|---|---|---|---|
| | FL | LR | *o.m.*[†] | FL | LR | *o.m.* | FL | LR | *o.m.* | FL | LR | *o.m.* |
| *Leaves* | | | | | | | | | | | | |
| MG_0 | 15.54 | 15.83 | *15.69 B* | 4.06 e | 5.14 d | *4.60* | 13.85 | 12.59 | *13.22 B* | 7.35 c | 6.50 d | *6.92* |
| MG_30 | 17.85 | 19.03 | *18.44 AB* | 5.05 d | 10.41 b | *7.73* | 16.75 | 16.32 | *16.54 A* | 13.05 c | 14.65 b | *13.85* |
| MG_60 | 20.15 | 19.65 | *19.90 A* | 8.64 c | 11.82 a | *10.23* | 16.19 | 18.45 | *17.32 A* | 17.22 a | 21.72 a | *19.47* |
| *o.m.* | *17.85* | *18.17* | | *5.92* | *9.12* | | *15.60* | *15.78* | | *12.54* | *1.429* | |
| *PAR availability* | *n.s.* | | | ***(0.013)* | | | *n.s.* | | | *n.s.* | | |
| *Mg rates* | ***(0.909)* | | | ***(0.026)* | | | ***(0.980)* | | | ***(0.048)* | | |
| *PAR × Mg* | *n.s.* | | | ***(0.036)* | | | *n.s.* | | | *** (0.068)* | | |
| *Storage root* | | | | | | | | | | | | |
| MG_0 | 10.49 | 11.53 | *11.01 B* | 2.16 d | 1.85 e | *2.01* | 10.47 | 10.18 | *10.32 B* | 2.06 d | 1.94 d | *2.00* |
| MG_30 | 12.50 | 12.09 | *12.29 A* | 3.04 b | 3.92 b | *3.48* | 12.09 | 10.76 | *11.43 A* | 2.89 b | 3.69 a | *3.29* |
| MG_60 | 13.08 | 12.61 | *12.85 A* | 2.78 a | 3.11 c | *2.94* | 12.12 | 10.95 | *11.54 A* | 2.53 c | 3.20 b | *2.86* |
| *o.m.* | *12.02* | *12.07* | | *2.66* | *2.96* | | *11.56 A* | *10.63 B* | | *2.49* | *2.94* | |
| *PAR availability* | *n.s.* | | | ***(0.002)* | | | ***(0.190)* | | | ***(0.005)* | | |
| *Mg rates* | **(0.590)* | | | ***(0.007)* | | | ***(0.301)* | | | ***(0.009)* | | |
| *PAR × Mg* | *n.s.* | | | ***(0.010)* | | | *n.s.* | | | ***(0.013)* | | |

*o.m.*[†]: overall means. * $p < 0.05$; ** $p < 0.01$; *n.s.* = not-significant. In brackets: standard error of differences between means (s.e.d.). Degrees of freedom: PAR availability. 1; Mg rates. 2; PAR availability × Mg rates. 2; Residual. 8.

**Table 3.** Chlorophyll content (µg mg$^{-1}$ FW) in leaves of red beet plants subjected to two levels of photosynthetically active radiation (PAR) availability (100% PAR availability: Full Light, FL; 50% PAR availability: Light Reduction, LR) and three magnesium (Mg) application rates (0 kg ha$^{-1}$ of Mg: MG_0; 30 kg ha$^{-1}$ of Mg: MG_30; 60 kg ha$^{-1}$ of Mg: MG_60) at 143, 358 and 714 Growing Degree Days (GDD,°C) after transplanting in 2017 and at 158, 376 and 724 GDD after transplanting in 2018. Measurements are average values of three replications. Means followed by different letters (upper case letters: main effects; lower case letters: effects of interaction) significantly differ (Fisher's LSD, $p \leq 0.05$).

| | | | | Chlorophyll Content (µg mg$^{-1}$ FW) | | | | | | | | | | | |
|---|---|---|---|---|---|---|---|---|---|---|---|---|---|---|---|
| **Year** | **GDD** | **Treatments** | **ChlA** | | | **ChlB** | | | **ChlTot** | | | **ChlA / ChlB** | | | | |
| **2017** | | | **FL** | **LR** | *o.m.*† | **FL** | **LR** | *o.m.* | **FL** | **LR** | *o.m.* | **FL** | **LR** | *o.m.* | | |
| | 143 | MG_0 | 0.58 c | 0.73 c | *0.65* | 0.29 c | 0.30 c | *0.29* | 0.87 c | 1.03 c | *0.95* | 2.01 c | 2.42 c | *2.22* | | |
| | | MG_30 | 1.80 b | 3.10 a | *2.45* | 0.50 b | 0.61 a | *0.56* | 2.30 b | 3.71 a | *3.01* | 3.58 b | 5.08 a | *4.33* | | |
| | | MG_60 | 1.95 b | 2.96 a | *2.46* | 0.59 a | 0.59 a | *0.59* | 2.54 b | 3.55 a | *3.05* | 3.29 b | 5.06 a | *4.17* | | |
| | | *o.m.* | *1.44* | *2.26* | | *0.46* | *0.50* | | *1.90* | *2.76* | | *2.96* | *4.18* | | | |
| | | PAR availability | | **(0.071)* | | | **(0.008)* | | | **(0.072)* | | | **(0.147)* | | | |
| | | Mg rates | | **(0.087)* | | | **(0.014)* | | | **(0.083)* | | | **(0.220)* | | | |
| | | PAR × Mg | | **(0.123)* | | | **(0.020)* | | | **(0.118)* | | | *(0.311)* | | | |
| | 358 | MG_0 | 0.75 c | 0.80 c | *0.77* | 0.32 c | 0.45 c | *0.38* | 1.07 c | 1.24 c | *1.15* | 2.40 b | 1.79 c | *2.09* | | |
| | | MG_30 | 1.14 b | 2.54 a | *1.84* | 0.49 b | 0.80 a | *0.65* | 1.63 b | 3.34 a | *2.48* | 2.31 b | 3.18 a | *2.75* | | |
| | | MG_60 | 1.16 b | 2.66 a | *1.91* | 0.62 b | 0.87 a | *0.74* | 1.78 b | 3.53 a | *2.65* | 1.88 c | 3.08 a | *2.48* | | |
| | | *o.m.* | *1.01* | *2.00* | | *0.48* | *0.70* | | *1.49* | *2.70* | | *2.19* | *2.68* | | | |
| | | PAR availability | | **(0.059)* | | | **(0.012)* | | | **(0.052)* | | | *(0.119)* | | | |
| | | Mg rates | | **(0.071)* | | | **(0.029)* | | | **(0.087)* | | | **(0.144)* | | | |
| | | PAR × Mg | | **(0.101)* | | | *(0.041)* | | | **(0.124)* | | | **(0.203)* | | | |
| | 714 | MG_0 | 0.80 d | 0.66 d | *0.73* | 0.31 e | 0.41 d | *0.36* | 1.11 d | 1.08 d | *1.09* | 2.56 ab | 1.63 d | *2.09* | | |
| | | MG_30 | 1.25 c | 2.51 a | *1.88* | 0.55 c | 0.92 a | *0.74* | 1.80 c | 3.43 a | *2.62* | 2.27 bc | 2.74 a | *2.51* | | |
| | | MG_60 | 1.55 b | 2.58 a | *2.06* | 0.79 b | 0.98 a | *0.88* | 2.34 b | 3.56 a | *2.95* | 1.96 cd | 2.64 ab | *2.30* | | |
| | | *o.m.* | *1.20* | *1.92* | | *0.55* | *0.77* | | *1.75* | *2.69* | | *2.26* | *2.34* | | | |
| | | PAR availability | | **(0.058)* | | | **(0.015)* | | | **(0.064)* | | | *n.s.* | | | |
| | | Mg rates | | **(0.053)* | | | **(0.027)* | | | **(0.063)* | | | *(0.122)* | | | |
| | | PAR × Mg | | **(0.076)* | | | **(0.039)* | | | **(0.090)* | | | ** (0.173)* | | | |

**Table 3.** *Cont.*

| | | | Chlorophyll Content (µg mg$^{-1}$ FW) | | | | | | | | | | | |
|---|---|---|---|---|---|---|---|---|---|---|---|---|---|---|
| Year | GDD | Treatments | ChlA | | | ChlB | | | ChlTot | | | ChlA / ChlB | | |
| **2017** | | | FL | LR | *o.m.*† | FL | LR | *o.m.* | FL | LR | *o.m.* | FL | LR | *o.m.* |
| 2018 | 158 | MG_0 | 0.44 c | 0.53 c | *0.48* | 0.23 | 0.26 | *0.24 B* | 0.67 | 0.78 | 0.73 b | 1.92 b | 2.07 b | *1.99* |
| | | MG_30 | 1.46 b | 2.78 a | *2.12* | 0.76 | 0.67 | *0.71 A* | 2.22 | 3.45 | 2.83 a | 1.95 b | 4.17 a | *3.06* |
| | | MG_60 | 1.30 b | 2.65 a | *1.97* | 0.55 | 0.59 | *0.57 A* | 1.85 | 3.24 | 2.54 a | 2.35 b | 4.63 a | *3.49* |
| | | *o.m.* | *1.07* | *1.99* | | *0.51* | *0.50* | | *1.58 A* | *2.49 B* | | *2.07* | *3.63* | |
| | | *PAR availability* | **(0.112) | | | | n.s. | | ** (0.140) | | | **(0.579) | | |
| | | *Mg rates* | **(0.228) | | | | **(0.065) | | **(0.285) | | | **(0.521) | | |
| | | *PAR × Mg* | *(0.323) | | | | n.s. | | n.s. | | | **(0.737) | | |
| | 376 | MG_0 | 0.55 | 0.65 | *0.60 B* | 0.30 | 0.47 | *0.38 B* | 0.85 b | 1.11 b | *0.98* | 1.80 | 1.41 | *1.60 B* |
| | | MG_30 | 1.02 | 2.36 | *1.69 A* | 0.44 | 1.17 | *0.80 A* | 1.46 b | 3.53 a | *2.49* | 2.32 | 2.14 | *2.23 AB* |
| | | MG_60 | 1.13 | 2.73 | *1.93 A* | 0.55 | 0.81 | *0.68 AB* | 1.68 b | 3.53 a | *2.60* | 2.08 | 3.48 | *2.78 A* |
| | | *o.m.* | *0.90 B* | *1.91 A* | | *0.43 B* | *0.81 A* | | *1.33* | *2.72* | | *2.07* | *2.34* | |
| | | *PAR availability* | ** (0.141) | | | | ** (0.078) | | **(0.209) | | | *(0.078) | | |
| | | *Mg rates* | **(0.278) | | | | *(0.112) | | **(0.301) | | | *(0.389) | | |
| | | *PAR × Mg* | n.s. | | | | n.s. | | *(0.426) | | | n.s. | | |
| | 724 | MG_0 | 0.52 d | 0.59 d | *0.56* | 0.23 d | 0.32 d | *0.28* | 0.75 d | 0.91 d | *0.83* | 2.35 ab | 1.82 c | *2.08* |
| | | MG_30 | 1.21 c | 2.34 a | *1.77* | 0.50 c | 1.15 a | *0.82* | 1.71 c | 3.48 a | *2.60* | 2.45 a | 2.04 bc | *2.24* |
| | | MG_60 | 1.62 b | 2.40 a | *2.01* | 0.85 b | 0.92 b | *0.89* | 2.47 b | 3.32 a | *2.90* | 1.90 c | 2.64 a | *2.27* |
| | | *o.m.* | *1.12* | *1.78* | | *0.53* | *0.80* | | *1.64* | *2.57* | | *2.23* | *2.17* | |
| | | *PAR availability* | **(0.060) | | | | **(0.047) | | **(0.105) | | | n.s. | | |
| | | *Mg rates* | **(0.054) | | | | **(0.035) | | **(0.079) | | | n.s. | | |
| | | *PAR × Mg* | **(0.076) | | | | **(0.050) | | **(0.111) | | | **(0.173) | | |

*o.m.*†: overall means. * $p < 0.05$; ** $p < 0.01$; *n.s.* = not-significant. In brackets: standard error of differences between means (s.e.d.). Degrees of freedom: PAR availability, 1; Mg rates, 2; PAR availability × Mg rates, 2; Residual, 8.

**Table 4.** Normalized Difference Vegetation Index (NDVI$_{670}$), Photochemical Reflectance Index (PRI), Chlorophyll Index (CI) and Soil Plant Analysis Development (SPAD) as recorded for red beet plants subjected to two levels of photosynthetically active radiation (PAR) availability (100% PAR availability: Full Light, FL; 50% PAR availability: Light Reduction, LR) and three magnesium (Mg) application rates (0 kg ha$^{-1}$ of Mg: MG_0; 30 kg ha$^{-1}$ of Mg: MG_30; 60 kg ha$^{-1}$ of Mg: MG_60) at 143, 358 and 714 Growing Degree Days (GDD,°C) after transplanting in 2017 and at 158, 376 and 724 GDD after transplanting in 2018. Measurements are average values of three replications. Means followed by different letters (upper case letters: main effects; lower case letters: effects of interaction) significantly differ (Fisher's LSD, $p \leq 0.05$).

| Year | GDD | Treatments | NDVI$_{670}$ | | | PRI | | | CI | | | SPAD | | |
|---|---|---|---|---|---|---|---|---|---|---|---|---|---|---|
| **2017** | | | FL | LR | o.m.† | FL | LR | o.m. | FL | LR | o.m. | FL | LR | o.m. |
| | 143 | MG_0 | 0.366 | 0.333 | 0.350 B | −0.042 | −0.042 | −0.042 B | 0.165 | 0.179 | 0.172 B | 40.6 c | 39.9 c | 40.3 |
| | | MG_30 | 0.443 | 0.459 | 0.451 AB | −0.029 | −0.024 | −0.026 A | 0.252 | 0.255 | 0.254 AB | 48.9 b | 54.7 a | 51.8 |
| | | MG_60 | 0.573 | 0.513 | 0.543 A | −0.024 | −0.030 | −0.027 A | 0.333 | 0.286 | 0.310 A | 47.1 b | 53.2 a | 50.2 |
| | | o.m. | 0.461 | 0.435 | | −0.031 | −0.032 | | 0.250 | 0.240 | | 45.5 | 49.3 | |
| | | PAR availability | | n.s. | | | n.s. | | | n.s. | | | *(0.990) | |
| | | Mg rates | | **(0.045) | | | *(0.005) | | | **(0.029) | | | ** (1.271) | |
| | | PAR × Mg | | n.s. | | | n.s. | | | n.s. | | | *(1.798) | |
| | 358 | MG_0 | 0.393 | 0.434 | 0.414 | −0.058 | −0.048 | −0.053 | 0.155 | 0.188 | 0.171 | 35.2 | 39.6 | 37.4 B |
| | | MG_30 | 0.448 | 0.466 | 0.457 | −0.055 | −0.041 | −0.048 | 0.210 | 0.229 | 0.219 | 45.5 | 54.8 | 50.2 A |
| | | MG_60 | 0.464 | 0.524 | 0.494 | −0.048 | −0.039 | −0.044 | 0.238 | 0.279 | 0.259 | 48.7 | 55.2 | 52.0 A |
| | | o.m. | 0.435 | 0.475 | | −0.054 A | −0.043 B | | 0.201 | 0.232 | | 43.1 B | 49.9 A | |
| | | PAR availability | | n.s. | | | *(0.003) | | | n.s. | | | ** (1.156) | |
| | | Mg rates | | n.s. | | | n.s. | | | n.s. | | | ** (1.639) | |
| | | PAR × Mg | | n.s. | | | n.s. | | | n.s. | | | n.s. | |
| | 714 | MG_0 | 0.564 | 0.576 | 0.570 AB | −0.044 | −0.031 | −0.037 B | 0.270 | 0.295 | 0.282 B | 46.3 | 41.5 | 43.9 B |
| | | MG_30 | 0.481 | 0.531 | 0.506 B | −0.044 | −0.032 | −0.038 B | 0.241 | 0.267 | 0.254 B | 52.0 | 52.0 | 52.0 A |
| | | MG_60 | 0.616 | 0.652 | 0.634 A | −0.036 | −0.027 | −0.031 A | 0.326 | 0.331 | 0.328 A | 48.9 | 55.7 | 52.3 A |
| | | o.m. | 0.554 | 0.586 | | −0.041 B | −0.030 A | | 0.279 | 0.298 | | 49.1 | 49.8 | |
| | | PAR availability | | n.s. | | | **(0.001) | | | n.s. | | | n.s. | |
| | | Mg rates | | **(0.027) | | | **(0.001) | | | **(0.029) | | | ** (2.090) | |
| | | PAR × Mg | | n.s. | | | n.s. | | | n.s. | | | n.s. | |

**Table 4.** *Cont.*

| Year | GDD | Treatments | NDVI$_{670}$ | | | PRI | | | CI | | | SPAD | | |
|---|---|---|---|---|---|---|---|---|---|---|---|---|---|---|
| **2017** | | | **FL** | **LR** | ***o.m.***$^{†}$ | **FL** | **LR** | ***o.m.*** | **FL** | **LR** | ***o.m.*** | **FL** | **LR** | ***o.m.*** |
| 2018 | 158 | MG_0 | 0.423 | 0.475 | *0.449* B | −0.012 | 0.000 | *−0.006* | 0.140 | 0.132 | *0.136* B | 24.4 d | 29.4 c | *25.9* |
| | | MG_30 | 0.592 | 0.617 | *0.604* A | −0.006 | −0.002 | *−0.004* | 0.193 | 0.180 | *0.187* A | 31.5 c | 36.4 b | *34.0* |
| | | MG_60 | 0.564 | 0.672 | *0.618* A | −0.014 | −0.005 | *−0.010* | 0.188 | 0.226 | *0.207* A | 35.9 b | 47.8 a | *41.9* |
| | | *o.m.* | 0.526 B | *0.588* A | | −0.010 | −0.002 | | 0.174 | *0.179* | | 29.9 | *37.9* | |
| | | *PAR availability* | | *(0.020) | | | *n.s.* | | | *n.s.* | | | **(0.945) | |
| | | *Mg rates* | | **(0.022) | | | *n.s.* | | | **(0.014) | | | **(1.160) | |
| | | *PAR × Mg* | | *n.s.* | | | *n.s.* | | | *n.s.* | | | **(1.640) | |
| | 376 | MG_0 | 0.569 | 0.567 | *0.568* | −0.026 | −0.024 | *−0.025* | 0.315 | 0.290 | *0.303* | 31.0 d | 31.6 d | *31.3* |
| | | MG_30 | 0.658 | 0.530 | *0.594* | −0.025 | −0.028 | *−0.026* | 0.335 | 0.250 | *0.292* | 33.7 cd | 39.0 b | *36.4* |
| | | MG_60 | 0.565 | 0.704 | *0.634* | −0.026 | −0.010 | *−0.018* | 0.289 | 0.353 | *0.321* | 36.0 bc | 50.7 a | *43.4* |
| | | *o.m.* | *0.597* | *0.600* | | −0.025 | −0.020 | | 0.313 | *0.298* | | 33.6 | *40.5* | *37.0* |
| | | *PAR availability* | | *n.s.* | | | *n.s.* | | | *n.s.* | | | **(1.055) | |
| | | *Mg rates* | | *n.s.* | | | *n.s.* | | | *n.s.* | | | **(0.964) | |
| | | *PAR × Mg* | | *n.s.* | | | *n.s.* | | | *n.s.* | | | **(1.364) | |
| | 724 | MG_0 | 0.461 | 0.465 | *0.463* B | −0.037 | −0.041 | *−0.039* B | 0.220 | 0.218 | *0.219* B | 30.6 e | 39.3 c | *35.0* |
| | | MG_30 | 0.625 | 0.672 | *0.649* A | −0.015 | −0.018 | *−0.016* A | 0.337 | 0.330 | *0.334* A | 35.2 d | 48.9 b | *42.1* |
| | | MG_60 | 0.605 | 0.709 | *0.657* A | −0.018 | −0.011 | *−0.014* A | 0.323 | 0.393 | *0.358* A | 37.4 d | 51.8 a | *44.6* |
| | | *o.m.* | *0.564* | *0.615* | | −0.023 | −0.023 | | 0.293 | *0.314* | | 34.4 | *46.7* | |
| | | *PAR availability* | | *n.s.* | | | *n.s.* | | | *n.s.* | | | **(0.471) | |
| | | *Mg rates* | | **(0.025) | | | **(0.003) | | | **(0.023) | | | **(0.604) | |
| | | *PAR × Mg* | | *n.s.* | | | *n.s.* | | | *n.s.* | | | **(0.854) | |

*o.m.*$^{†}$: overall means. * $p < 0.05$; ** $p < 0.01$; *n.s.* = not-significant. In brackets: standard error of differences between means (s.e.d.). Degrees of freedom: PAR availability, 1; Mg rates, 2; PAR availability × Mg rates, 2; Residual.

### 3.2. Nitrogen and Magnesium Accumulation and Chlorophyll Content

N concentration in plant tissues was strongly influenced by Mg rather than PAR availability (Table 2), with highly significant linear correlations between Mg rates and N concentration (leaves: $R^2 = 0.99$ and 0.91 in 2017 and 2018, respectively; storage root: $R^2 = 0.97$ and 0.91 in 2017 and 2018, respectively; data not shown). The highest values were recorded for MG_60 treatments (in 2017: 19.90 and 12.85 mg g$^{-1}$ DW for leaves and storage root, respectively, averaged over PAR availability; in 2018: 17.32 and 11.54 mg g$^{-1}$ DW for leaves and storage root, respectively, averaged over PAR availability).

As expected, Mg concentration in leaves was stimulated by Mg application rates and increased by 87% in MG_30 and 157% in MG_60 with respect to MG_0 treatment (averaged over years and PAR availability; Table 2). Besides, LR induced values of 54% and 19% higher than FL in 2017 and 2018, respectively (Table 2). The highest Mg concentration values were indeed obtained with "LR × MG_60" (Table 2).

A similar behavior was also registered in both seasons for storage roots, although the highest values were reached by the combination "LR × MG_30" (3.92 mg g$^{-1}$ DW and 3.69 mg g$^{-1}$ DW in 2017 and 2018, respectively; Table 2).

The contents of ChlA, ChlB and ChlTot (ChlA + ChlB) in leaves are reported in Table 3. ChlTot increased in response to both Mg application and shading conditions, due to the contemporary increase of ChlA and ChlB. LR gave the highest ChlA and ChlB values (Table 3) as well as, following the application of MG_60, ChlA raised up by 203% and 264%, while ChlB by 114% and 145% in 2017 and 2018, respectively (Table 3). Consequently, the highest ChlTot values were generally observed for "LR × MG_60" (Table 3). The ChlA/ChlB trend was pretty unclear since the significance of treatments was influenced both by year and GDD. However, higher values were generally achieved for "LR × MG_60" (Table 3).

The indices estimating Chl content support data obtained with analytical determination (Table 4). "LR × MG_60" obtained higher SPAD, NDVI$_{670}$, PRI and CI values. Despite no significant effects being observed, LR exhibited higher NDVI$_{670}$ and CI values with respect to FL treatments (Table 4). Similar trends were observed for PRI, although the effects of treatments differed between GDD and growing seasons (Table 4).

### 3.3. Sugars

Total sugars (sucrose + glucose + fructose) in leaves at harvesting ranged from 3.97 g 100 g$^{-1}$ to 6.43 g 100 g$^{-1}$ in 2017 and from 3.93 g 100 g$^{-1}$ to 6.40 g 100 g$^{-1}$ in 2018 (Table 5). Regardless of years and PAR availability, Mg availability significantly reduced the total sugars content.

Sucrose was the most abundant sugar (2.85 versus 1.30 and 0.76 g 100 g$^{-1}$, on average, of sucrose, glucose and fructose, respectively), particularly under the "FL × MG_0" combination; fructose followed a similar behavior (Table 5). Under LR, higher glucose accumulations in red beet leaves were registered (averaging over Mg rates: 1.68 versus 1.19 g 100 g$^{-1}$ in LR and FL, respectively, in 2017; 1.36 versus 1.01 g 100 g$^{-1}$ in LR and FL, respectively, in 2018; Table 5).

In the storage root, total sugars ranged from 49.77 to 57.80 g 100 g$^{-1}$ in 2017 and from 42.10 to 56.47 g 100 g$^{-1}$ in 2018 (Table 5). Conversely to leaves, supplying Mg to plants significantly enhanced the total sugar content (on average, 56.25 and 52.30 g 100 g$^{-1}$ for MG_60 in 2017 and 2018, respectively; Table 5). Sucrose was confirmed as the most abundant sugar, whereas fructose and glucose were found only in small amounts (50.63 versus 0.76 and 0.30 g 100 g$^{-1}$, on average, of sucrose, glucose and fructose, respectively; Table 5).

**Table 5.** Content of individual sugars (g 100 g$^{-1}$ DW) in leaves and storage root of red beet plants subjected to two levels of photosynthetically active radiation (PAR) availability (100% PAR availability: Full Light, FL; 50% PAR availability: Light Reduction, LR) and three magnesium (Mg) application rates (0 kg ha$^{-1}$ of Mg: MG_0; 30 kg ha$^{-1}$ of Mg: MG_30; 60 kg ha$^{-1}$ of Mg: MG_60) at maturity in 2017 and 2018. Measurements are average values of three replications. Means followed by different letters (upper case letters: main effects; lower case letters: effects of interaction) significantly differ (Fisher's LSD, $p \leq 0.05$).

| | | | Sucrose | | | Glucose | | | Fructose | | | Total Sugars | | |
|---|---|---|---|---|---|---|---|---|---|---|---|---|---|---|
| Year | Treatments | | FL | LR | o.m.$^{\dagger}$ | FL | LR | o.m. | FL | LR | o.m. | FL | LR | o.m. |
| **2017** | | | | | | | | | | | | | | |
| | *Leaves* | MG_0 | 4.03 | 3.57 | *3.80 A* | 1.37 c | 1.43 bc | *1.40* | 0.97 | 1.03 | *1.00 A* | 6.43 | 5.97 | *6.20 A* |
| | | MG_30 | 2.43 | 3.33 | *2.88 AB* | 1.07 d | 1.70 ab | *1.38* | 0.70 | 0.73 | *0.72 B* | 4.20 | 5.77 | *4.98 B* |
| | | MG_60 | 2.27 | 1.87 | *2.07 B* | 1.07 d | 1.97 a | *1.52* | 0.63 | 0.80 | *0.72 B* | 3.97 | 4.63 | *4.30 B* |
| | | *o.m.* | *2.91* | *2.92* | | *1.19* | *1.68* | | *0.77* | *0.86* | | *4.87* | *5.46* | |
| | *PAR availability* | | | *n.s.* | | | *\*\*(0.183)* | | | *n.s.* | | | *n.s.* | |
| | *Mg rates* | | | *\*\*(0.358)* | | | *n.s.* | | | *\*(0.079)* | | | *\*\*(0.350)* | |
| | *PAR × Mg* | | | *n.s.* | | | *\*\*(0.025)* | | | *n.s.* | | | *n.s.* | |
| | *Storage root* | MG_0 | 48.70 | 52.27 | *50.48 B* | 0.83 | 0.73 | *0.78* | 0.23 | 0.13 | *0.18 B* | 49.77 | 53.13 | *51.45 B* |
| | | MG_30 | 52.20 | 54.33 | *53.27 AB* | 0.93 | 0.97 | *0.95* | 0.43 | 0.53 | *0.48 A* | 53.57 | 55.83 | *54.70 AB* |
| | | MG_60 | 56.57 | 53.23 | *54.90 A* | 0.87 | 1.00 | *0.93* | 0.37 | 0.47 | *0.42 A* | 57.80 | 54.70 | *56.25 A* |
| | | *o.m.* | *52.49* | *53.28* | | *0.88* | *0.90* | | *0.34* | *0.38* | | *53.71* | *54.56* | |
| | *PAR availability* | | | *n.s.* | | | *n.s.* | | | *n.s.* | | | *n.s.* | |
| | *Mg rates* | | | *\*(1.365)* | | | *n.s.* | | | *\*\*(0.060)* | | | *\*(1.365)* | |
| | *PAR × Mg* | | | *n.s.* | | | *n.s.* | | | *n.s.* | | | *n.s.* | |
| **2018** | | | | | | | | | | | | | | |
| | *Leaves* | MG_0 | 4.43 | 3.17 | *3.80 A* | 1.07 | 1.40 | *1.23* | 0.90 | 0.93 | *0.92 A* | 6.40 | 5.50 | *5.95 A* |
| | | MG_30 | 2.70 | 2.17 | *2.43 B* | 1.00 | 1.33 | *1.17* | 0.53 | 0.73 | *0.63 B* | 4.23 | 4.23 | *4.23 B* |
| | | MG_60 | 2.40 | 1.80 | *2.10 B* | 0.97 | 1.33 | *1.15* | 0.57 | 0.60 | *0.58 B* | 3.93 | 3.73 | *3.83 B* |
| | | *o.m.* | *3.18* | *2.38* | | *1.01 B* | *1.36 A* | | *0.67* | *0.76* | | *4.86* | *4.49* | |
| | *PAR availability* | | | *n.s.* | | | *\*(0.095)* | | | *n.s.* | | | *n.s.* | |
| | *Mg rates* | | | *\*\*(0.178)* | | | *n.s.* | | | *\*\*(0.077)* | | | *\*\*(0.229)* | |
| | *PAR × Mg* | | | *n.s.* | | | *n.s.* | | | *n.s.* | | | *n.s.* | |
| | *Storage root* | MG_0 | 41.10 | 41.53 | *41.32 B* | 0.67 ab | 0.60 ab | *0.63* | 0.33 | 0.20 | *0.27 A* | 42.10 | 42.33 | *42.22 B* |
| | | MG_30 | 55.40 | 49.20 | *52.30 A* | 0.77 a | 0.57 ab | *0.63* | 0.30 | 0.23 | *0.27 A* | 56.47 | 50.00 | *53.23 A* |
| | | MG_60 | 50.93 | 52.13 | *51.53 A* | 0.47 b | 0.73 a | *0.63* | 0.20 | 0.13 | *0.17 B* | 51.60 | 53.00 | *52.30 A* |
| | | *o.m.* | *49.14* | *47.62* | | *0.63* | *0.63* | | *0.28 A* | *0.19 B* | | *50.06* | *48.44* | |
| | *PAR availability* | | | *n.s.* | | | *n.s.* | | | *\*\*(0.015)* | | | *n.s.* | |
| | *Mg rates* | | | *\*(1.918)* | | | *n.s.* | | | *\*\*(0.025)* | | | *\*\*(1.879)* | |
| | *PAR × Mg* | | | *n.s.* | | | *\*(0.108)* | | | *n.s.* | | | *n.s.* | |

*o.m.*$^{\dagger}$: overall means. * $p < 0.05$; ** $p < 0.01$; *n.s.* = not-significant. In brackets: standard error of differences between means (s.e.d.). Degrees of freedom: PAR availability, 1; Mg rates, 2; PAR availability × Mg rates, 2; Residual, 8.

### 3.4. Polyphenols and Betalains Content

Total polyphenols and betalains content were investigated in storage roots at harvesting (Figure 1).

TPC was significantly affected both by PAR availability and Mg application (Figure 1a): MG_0 gave significantly higher TPC than MG_30 and MG_60 (averaged over PAR availability, 2017: 51.14 versus 42.67 and 41.80 mg GAE 100 $g^{-1}$ in MG_0, MG_30 and MG_60, respectively; 2018: 46.20 versus 36.03 and 39.03 mg GAE 100 $g^{-1}$ in MG_0, MG_30 and MG_60, respectively; Figure 1a). Besides, values basically decreased in response to LR conditions (2017: 50.54 versus 39.84 mg GAE 100 $g^{-1}$ in FL and LR, respectively; Figure 1a).

BC decreased significantly as Mg rates increased, especially under LR conditions (Figure 1b), as demonstrated by the highest value recorded for "FL x MG_0" (18.24 and 16.06 mg $g^{-1}$ in 2017 and 2018, respectively). Conversely, betaxanthins (BX) content increased significantly under LR by 17% and 27%, on average, in 2017 and 2018, respectively (Figure 1c). BC/BX ratios generally decreased under LR and with higher Mg application levels (Table 6); Mg deficiency allowed significantly higher values of BC + BX (MG_0, 28.67 and 25.79 mg $g^{-1}$, on average, in 2017 and 2018, respectively; Table 6).

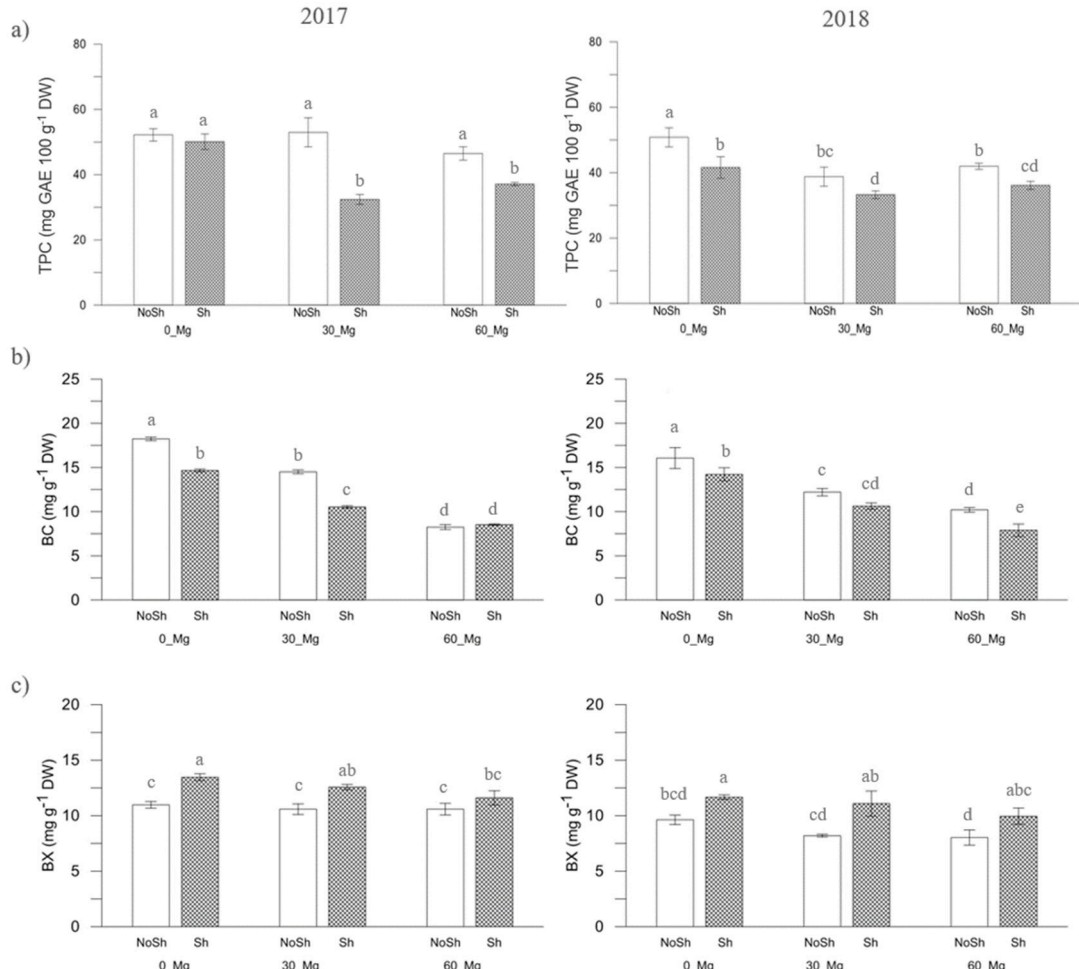

**Figure 1.** (**a**) Total polyphenols content (TPC, mg gallic acid equivalents (GAE) 100 $g^{-1}$ dry weight, DW); (**b**) betacyanins (BC, mg $g^{-1}$ DW) and (**c**) betaxanthins (BX, mg $g^{-1}$ DW) content in the storage roots of red beet plants subjected to two levels of photosynthetically active radiation (PAR) availability (100% PAR availability: Full Light, FL; 50% PAR availability: Light Reduction, LR) and three magnesium (Mg) application rates (0 kg $ha^{-1}$ of Mg: MG_0; 30 kg $ha^{-1}$ of Mg: MG_30; 60 kg $ha^{-1}$ of Mg: MG_60) during two growing cycles (2017 and 2018). Average values ± standard errors are depicted; *n* = 3 independent replicates. Different letters stand for statistically significantly differences at *p* ≤ 0.05 (Fisher's LSD test).

**Table 6.** Total betalains content expressed as the sum of betacyanins (BC, mg g$^{-1}$) and betaxanthins (BX, mg g$^{-1}$) (BC + BX) as well as the ratio of BC/BX as observed in the storage root of red beet plants subjected to two levels of photosynthetically active radiation (PAR) availability (100% PAR availability: Full Light, FL; 50% PAR availability: Light Reduction, LR) and three magnesium (Mg) application rates (0 kg ha$^{-1}$ of Mg: MG_0; 30 kg ha$^{-1}$ of Mg: MG_30; 60 kg ha$^{-1}$ of Mg: MG_60) at maturity in 2017 and 2018. Measurements are average values of three replications. Means followed by different letters (upper case letters: main effects; lower case letters: effects of interaction) significantly differ (Fisher's LSD, $p \leq 0.05$).

| Treatments | 2017 | | | | | | 2018 | | | | | |
| | BC + BX | | | BC/BX | | | BC + BX | | | BC/BX | | |
| | FL | LR | o.m.[†] | FL | LR | o.m. | FL | LR | o.m. | FL | LR | o.m. |
|---|---|---|---|---|---|---|---|---|---|---|---|---|
| MG_0 | 29.22 a | 28.12 a | *28.67* | 1.66 a | 1.09 c | *1.38* | 25.50 | 26.08 | *25.79 A* | 1.65 | 1.24 | *1.44 A* |
| MG_30 | 25.10 b | 23.10 c | *24.10* | 1.38 b | 0.84 d | *1.11* | 19.98 | 22.15 | *21.06 B* | 1.44 | 1.02 | *1.23 AB* |
| MG_60 | 18.84 d | 20.13 d | *19.49* | 0.79 d | 0.74 d | *0.76* | 17.20 | 18.88 | *18.04 C* | 1.15 | 0.91 | *1.03 B* |
| *o.m.* | *24.38* | *23.78* | | *1.28* | *0.89* | | *20.90* | *22.37* | | *1.41* | *1.06* | |
| PAR availability | | n.s. | | | **(0.043) | | | n.s. | | | n.s. | |
| Mg rates | | **(0.487) | | | **(0.032) | | | **(0.905) | | | **(0.066) | |
| PAR × Mg | | *(0.689) | | | **(0.046) | | | n.s. | | | n.s. | |

*o.m.*[†]: overall means. * $p < 0.05$; ** $p < 0.01$; *n.s.* = not-significant. In brackets: standard error of differences between means (s.e.d.). Degrees of freedom: PAR availability, 1; Mg rates, 2; PAR availability × Mg rates, 2; Residual, 8.

## 4. Discussion

Light and mineral nutrition are among the major determinants of biomass accumulation and partitioning in plants. In this study, Mg fertilization revealed a stronger effect on red beet growth than PAR availability, probably in view of the large number of Mg-requiring enzymes that are involved in energetic metabolism, allowing for complex responses [35]. We observed that Mg availability positively influences storage root DW, as already found in sugar beet [26]; it was partially expected, since a minimum threshold of Mg supply is required for maximizing the carbohydrate transport into sink organs, hence promoting high yields. Despite differences among species and environmental conditions (e.g. light intensity), Mg deficiency is suspected to be at the threshold value of Mg concentration in leaves of 2 mg g$^{-1}$ DM [36]. In our study, we never registered Mg concentrations at the Mg deficiency level, even under the MG_0 treatment. It could be attributable to (i) differences in plant genotype; (ii) sampling: we performed our analysis on the whole aerial biomass (i.e. including both old and young leaves) rather than on the uppermost, youngest leaves, which normally exhibit lower Mg contents [12]; (iii) differences in plant growth substrates: our potting soil, although containing Mg to a very low extent, could have guaranteed a certain Mg source.

Alteration in plant growth is mainly attributable to abnormal physiological processes reflected in impairments of photosynthetic $CO_2$ fixation [37] and photosynthesis, where Mg plays a key role through: (i) affecting the activity of enzymes involved in photosynthetic carbon metabolism, (ii) playing a major role in photosynthetic electron transport in photosystem I (PSI) and PSII reaction centers and, (iii) being both a structural component of Chl molecules and being needed for its biosynthesis [9]. Our work confirms that there is higher ChlA and ChlB concentration in leaves of red beet supplied with Mg [38–40] and that Mg deficiency symptoms (leaf yellowing as interveinal chlorosis in older leaves) are highly dependent on light intensity [7,9,41,42]. In our experience, sub-optimal Mg availability impairs photosynthetic reaction as also demonstrated by the higher total sugars contents in leaves, since a negative correlation between sugar levels, photosynthetic activities and chlorophyll content is generally reported [41]. However, such an increase (up to 1.8-fold) was lower than other literature results (even up to 12-fold) [41,43]. In general, sugar accumulation in Mg-deficient plant leaves occurs before any noticeable change in Chl content [41]; as a consequence, chlorosis cannot be considered a suitable tool for the early diagnosis of Mg deficiency [44]. To this purpose, the vegetation indices selected in our work (NDVI$_{670}$, CI and PRI) - validated at the canopy scale [45,46] - have demonstrated that it is good to estimate changes in Chl content (as indicated by the higher correlation values with total Chl content of 0.67, 0.60 and 0.65, on average, for NDVI$_{670}$, CI and PRI, respectively) early, regardless of the appearance of symptoms on the leaves. However, it is important to specify that in our experimental conditions, a slight yellowing of the oldest leaves represented the only symptom of sub-optimal Mg supply and occurred later on the crop cycle. Among the vegetation indices, NDVI$_{670}$ and CI performed well in both years, showing promising potential as mineral deficiency predictors, while PRI, despite the high correlation with total Chl content (coefficient of correlation 0.77 in 2017), was more sensible to year to year variations; further studies are therefore recommended.

The well-known direct functional and/or structural effect(s) of Mg on the phloemloading process of sucrose [9], have probably caused, under sub-optimal Mg levels, an impairment of phloem photosynthate export from source to sink organs [10], with significant influence on the quality traits of storage roots. Reducing Mg supply to plants proved to be an effective tool for enhancing secondary metabolites production, like polyphenols and betalains (BC and BX). In Mg-deficient leaves the under-utilization of absorbed light energy in the $CO_2$ fixation process, intensifies the flow of light energy, potentially addressed at reducing equivalents to molecular $O_2$ to form ROS [9]. Consequently, an accumulation of antioxidants (polyphenols and betalains) could be over-produced in the detoxification process. In addition, the subsequent transport via phloem to the storage organs is equally important as well as the "concentration effect" taking place due to reduced yield. Our work, indeed, demonstrated that the effects of PAR reduction in inhibiting the biosynthesis of secondary metabolites was noticeable in this species [17] and that the calibration of PAR/Mg availability is a promising tool to modify

the BC(red)/BX(yellow) ratio to values which translates into high quality products [47]. Also, high Mg availabilities are essential for the accumulation of macronutrients in the storage roots of red beet [10], including sugars [48] and N. For the latter, an inhibited metabolism was observed in spinach, probably due to the inhibition of the activities of glutamate synthase, glutamate dehydrogenase, glutamic-pyruvic transaminase, glutamic-oxaloace protease transaminase and urease [49].

## 5. Conclusions

Mg supply had stronger effects on regulating red beet growth and physiological traits rather than PAR availability. Plants reacted to a sub-optimal Mg rate, with a strong decrease in biomass accumulation, yield and partitioning of photosynthates from source to sink organs. Selected vegetation indices, NDVI$_{670}$ and CI, were very effective in the early estimation of the changes in Chl content in the red beet leaves; it follows that further studies are needed to assess their role in the early discrimination of Mg deficiency in this species.

The modification of the combination PAR/Mg nutrition induced significant variations of total polyphenol and betalains content as well as macronutrients, hence resulting in a intertesting tool to enhance the health and nutritional values of edible biomass.

**Author Contributions:** Conceptualization, S.DE. and A.G.; methodology, F.S. and A.G.; software, S.DE. and G.P.; validation, G.P. and S.DE.; formal analysis, A.G.; investigation, G.P. and S.DE.; resources, F.S. and M.P.; data curation, F.S. and A.G.; writing original draft preparation, S.DE., F.S. and A.G.; writing review and editing, M.P.; visualization, F.S., S.DE., G.P. and A.G.; supervision, M.P.

**Funding:** This research did not receive any specific grant from funding agencies in the public, commercial or not-for-profit sectors.

**Conflicts of Interest:** The authors declare no conflict of interest.

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
