# Peer review of "Yield, Quality and Physiological Traits of Red Beet Under Different Magnesium Nutrition and Light Intensity Levels"

_agronomy, doi:10.3390/agronomy9070379_

Round 1
Reviewer 1 Report
Introduction part
The introduction is well written and structured. However, it can substantially be improved by adding short information about the vegetation indices and the polyphenols. For example, what is the difference between NDVI, PRI and CI? Which valuable information do they convey? What are the functions of betacyanins and betaxanthins? Why should their contents be modified in edible plant tissues? Giving information on these aspects can prepare the reader much better for the presented results and discussion.
L49: CO2: 2 as subscript
L54: reddish spots do not occur in all plant species, but most likely in red beet. I suggest to specify here that reddish spots occur on red beet leaves.
L62-66: this sentence is very long. Why not splitting it in 2 sentences? That would be easier for the reader.
Materials and Methods
The Materials and Methods part is well elaborated, but the following comments should be considered and some minor corrections should be done:
L88: the PAR intensities must be given. Please add this important information.
L91: “consisted of”, not “consisted on”
L101: 12.6 %: delete the gap before %
L105: “consisted of”, not “consisted on”
L112: “consisted of”, not “consisted on”
L112: This part not clear. There were 6 treatments: Mg_0 +FL and +LR, Mg_30 +FL and +LR, and Mg_60 +FL and +LR. All of those treatments were replicated 3 times. This is also described in L89. This means 18 pots in total and 3 pots per treatment. It is not clear why one replication consists of 30 pots (L112) and why one treatment has 90 pots (L113). Please revise this paragraph.
L112: is the information of 5 rows necessary? It can be important to mention if, for example conditions in the greenhouse differ between the rows. But then, it should be indicated how the treatments were allocated. Completely randomized or in block, etc.?
L135: no need for the abbreviation “NIR” as it is not used in the further text
L139: “Normalized” and ”Vegetation” are in italics. I suppose they should not be in italics.
L159: a gap between g-1 and DW is missing
L171: please add the information on whether the leaf was a mature or a young leaf.
L178: a gap between g-1 and DW is missing
Results
The results part is clearly structured and kept to an adequate length. Some text editing has to be done (see comments below). Considering the tables: the tables seem to be another version than those submitted as Word documents. Hence, there should be some changes done (see also comments below). Secondly, the tables 3 and 4 are very complex. The authors can think of displaying the results in line graphs showing the changes over the time sequence. It can substantially improve understanding the numerous data which are interesting and worthwhile to present, but just not easy to grasp from a big table.
Comments to Table 1: on my screen, it seems like some parts are in bold and some not. However, the table provided in the word document looks fine. The figure caption in the word document is correct while in the manuscript it is not. Please cross check for L232: delete the dot after Mg: ; L232: part is in bold ; L234: “Means followed by different…” is doubled ; meaning of “o.m.” is not indicated.
Comment to Table 1: please add the information about what lower case und upper case letters mean with respect to the ANOVA.
L244/245: please replace the comma in the values with a dot
Comment to Table 2: on my screen, it seems like some parts are in bold and some not. However, the table provided in the word document looks fine. Please cross check for: meaning of “o.m.” is not indicated; replace the commas in the values with a dot; include a gap between g-1 and DW.
Comment to Table 2: do you show N content in mg plant-1 or N concentration in mg g-1 DW? Please cross check the table caption and the table header and correct.
L256: include a gap between Table and 2
L260: include a gap between g-1 and DW
L266: %%
L267: Results of ChlA/ChlB are not mentioned.
Comment to Table 3: again, it appears not uniform on my screen. Please cross check the Word and manuscript version.
Comment to Table 3: GDD has the superscript “a”, but its meaning is not explained
Comments to Table 4: again, the tables in the Word document and the manuscript are not identical. Please cross check thoroughly.
L289: total instead of Total
L292: “Conversely, LR allowed higher glucose….” This sentence is not clear. What do you want to state here? Please rephrase to make your statement more precise and clear.
L295: total instead of Total
L309: Figure 1 is included in the manuscript and supplementary files, hence data of TPC are not presented. Please either delete the respective part in 3.4. or include Fig 1.
L316-319: As Fig. 1 is missing, BC and BX data are not displayed separately. Please revise this section.
Discussion
L351: “like shading conditions..”: this is not clear. Please rephrase the sentence. Shading does not only increase glucose, but also fructose concentrations. While glucose and fructose are increased under shading, sucrose is decreased. This is an interesting finding and possible reasons for this phenomenon should be discussed.
L353-357: sentence is very long. Please split into 2 sentences.
L355-357: the strength of this part of discussion would be improved if you mention when first visible symptoms appeared and whether it was before or after changes in vegetation indices were measured. If data are available, please include them. You proved strongly with all chlorophyll measurements (ChlTot, NDVI, PRI, CI, SPAD) a decrease due to low Mg supply already at 143 GDD. It is interesting to know whether symptoms were already visible by then. This could also underline your conclusion that vegetation indices serve as good estimates for Mg deficiency.
L358: induce instead if induced
L359: probably due to… Please add “due”
L361: here is a contradiction to what was written above in L337: the Mg threshold is 2 mg g, but Mg concentration in the leaves are far above that threshold. Hence, the term “moderate controlled Mg deficiency” is not accurately used. There was no deficiency in the study. It would be an idea to include optimal Mg ranges for red beet, if available.
L373: “…resulted modified…” What is meant here? The N-metabolism was modified? Please rephrase.
Conclusions
L378: “led” instead of “leaded”
Author Response
Point by point response to Reviewer 1:
Introduction part
Rev comment: “The introduction is well written and structured. However, it can substantially be improved by adding short information about the vegetation indices and the polyphenols. For example, what is the difference between NDVI, PRI and CI? Which valuable information do they convey? What are the functions of betacyanins and betaxanthins? Why should their contents be modified in edible plant tissues? Giving information on these aspects can prepare the reader much better for the presented results and discussion.”
Our response: We have been added additional information on vegetation indices (VIs), highlighting also the differences between the selected VIs. At the same time, we have deepened the part related to betalains content as well as the importance of increasing their contents in plant tissues.
L49: CO2: 2 as subscript à now Line49: 2 has been written as subscript
L54: reddish spots do not occur in all plant species, but most likely in red beet. I suggest to specify here that reddish spots occur on red beet leaves. à now Lines 54-55: we have followed the suggestion made by Reviewer.
L62-66: this sentence is very long. Why not splitting it in 2 sentences? That would be easier for the reader. à we have followed the suggestion made by Reviewer.
Materials and Methods
The Materials and Methods part is well elaborated, but the following comments should be considered, and some minor corrections should be done:
L88: the PAR intensities must be given. Please add this important information à now Lines 109-110: we have added the PAR intensities as suggested.
L91: “consisted of”, not “consisted on” à now Line107 “consisted of” has been replaced with “consisted on”
L101: 12.6 %: delete the gap before % à now Line118: the suggestions have been followed
L105: “consisted of”, not “consisted on” à now Line123: “consisted of” has been replaced with “consisted on”
Rev comment: “L112: This part not clear. There were 6 treatments: Mg_0 +FL and +LR, Mg_30 +FL and +LR, and Mg_60 +FL and +LR. All of those treatments were replicated 3 times. This is also described in L89. This means 18 pots in total and 3 pots per treatment. It is not clear why one replication consists of 30 pots (L112) and why one treatment has 90 pots (L113). Please revise this paragraph.”
Our response: it is correct for us. Each experimental unit (one replication for each treatment) consisted on 30 pots and not on only 1 pot. Indeed, as specified in line 103, each pot hosted a single plant, so that 1 plant was not sufficient to obtain morphological, physiological and quality data (destructive samplings). Anyway, we have deleted all the sentence as specified in response to the following comment.
Rev comment: “L112: is the information of 5 rows necessary? It can be important to mention if, for example conditions in the greenhouse differ between the rows. But then, it should be indicated how the treatments were allocated. Completely randomized or in block, etc.?”
Our response: the information on 5 rows and 6 plants per row has been deleted as suggested. The experiment was arranged on a split-plot design with three replications, as reported at the beginning of the 2.1 paragraph. The sentence has been deleted and the information regarding the number of pots per replication has been added in line 104.
L135: no need for the abbreviation “NIR” as it is not used in the further text à the suggestions have been followed
L139: “Normalized” and ”Vegetation” are in italics. I suppose they should not be in italics à the suggestions have been followed
L159: a gap between g-1 and DW is missing à the suggestions have been followed
L171: please add the information on whether the leaf was a mature or a young leaf à now Lines 187-188: Despite samplings have been performed during crop cycle (at different GDD after transplanting), we always selected a fully expanded leaf or the more expanded leaf (early in the crop cycle), as specified in the text
L178: a gap between g-1 and DW is missing à the suggestions have been followed
Results
Rev comment: “The results part is clearly structured and kept to an adequate length. Some text editing has to be done (see comments below). Considering the tables: the tables seem to be another version than those submitted as Word documents. Hence, there should be some changes done (see also comments below). Secondly, the tables 3 and 4 are very complex. The authors can think of displaying the results in line graphs showing the changes over the time sequence. It can substantially improve understanding the numerous data which are interesting and worthwhile to present, but just not easy to grasp from a big table.”
Our response: We have followed your suggestions regarding the transformation of the tables 3 and 4 to graphs. As you can see (attached graph – only on Chl data as an example) the lines are very confused due to the high number of variables and experimental treatments. Moreover, it is difficult to see the effective significance of treatments starting from the s.e.d. values from ANOVA (reported as vertical bars for each GDD). Although we generally prefer data shown in figures, in this case, tables look better. What do you suggest?
Rev comment on Table 1: “on my screen, it seems like some parts are in bold and some not. However, the table provided in the word document looks fine. The figure caption in the word document is correct while in the manuscript it is not. Please cross check for L232: delete the dot after Mg: ; L232: part is in bold ; L234: “Means followed by different…” is doubled ; meaning of “o.m.” is not indicated. Please add the information about what lower case und upper case letters mean with respect to the ANOVA.”
Our response: All the suggestions in Table 1 have been followed.
L244/245: please replace the comma in the values with a dot à now Lines 264-265: the suggestions have been followed
Rev comment on Table 2: “on my screen, it seems like some parts are in bold and some not. However, the table provided in the word document looks fine. Please cross check for: meaning of “o.m.” is not indicated; replace the commas in the values with a dot; include a gap between g-1 and DW. Do you show N content in mg plant-1 or N concentration in mg g-1 DW? Please cross check the table caption and the table header and correct.”
Our response: All the suggestions in Table 2 have been followed.
L256: include a gap between Table and 2 à the suggestions have been followed
L260: include a gap between g-1 and DW à the suggestions have been followed
L266: %% à the suggestions have been followed
L267: Results of ChlA/ChlB are not mentioned à now Lines 286-288: results on ChlA: ChlB ratio have been added
Rev comment on Table 3: “again, it appears not uniform on my screen. Please cross check the Word and manuscript version - GDD has the superscript “a”, but its meaning is not explained”
Our response: All the suggestions on Table 3 have been followed; moreover, GDD meaning has been explained in the caption.
Rev comment on Table 4: “again, the tables in the Word document and the manuscript are not identical. Please cross check thoroughly.”
Our response: All the suggestions on Table 4 have been followed
L289: total instead of Total à the suggestions have been followed
Rev comment “L292: “Conversely, LR allowed higher glucose….” This sentence is not clear. What do you want to state here? Please rephrase to make your statement more precise and clear.”
Our response: the sentence has been rephrased (now Lines 316-318).
L295: total instead of Total à the suggestions have been followed
L309: Figure 1 is included in the manuscript and supplementary files, hence data of TPC are not presented. Please either delete the respective part in 3.4. or include Fig 1 (and “L316-319: As Fig. 1 is missing, BC and BX data are not displayed separately. Please revise this section.”) à we are sorry for the inconvenience. Figure 1 has been uploaded.
Discussion
Rev comment: “L351: “like shading conditions.”: this is not clear. Please rephrase the sentence. Shading does not only increase glucose, but also fructose concentrations. While glucose and fructose are increased under shading, sucrose is decreased. This is an interesting finding and possible reasons for this phenomenon should be discussed.”
Our response: We agree with the reviewer. Anyway, in this sentence we refer to sugar concentration in leaves, and not in storage organs, only in response to Mg availability. The sentence has been modified to gain a better understanding to readers.
L353-357: sentence is very long. Please split into 2 sentences. à the suggestions have been followed
Rev comment: “L355-357: the strength of this part of discussion would be improved if you mention when first visible symptoms appeared and whether it was before or after changes in vegetation indices were measured. If data are available, please include them. You proved strongly with all chlorophyll measurements (ChlTot, NDVI, PRI, CI, SPAD) a decrease due to low Mg supply already at 143 GDD. It is interesting to know whether symptoms were already visible by then. This could also underline your conclusion that vegetation indices serve as good estimates for Mg deficiency.”
Our response: The precise timing of symptoms appearance (in GDD) are not available. However, the beginning of leaf yellowing towards the leaf apex (which not included reddish spots on the leaf blade - also considering the threshold value of our Mg concentration in leaves) occurred later in the crop cycle. This statement has been included in the discussions, as suggested. Conclusions on vegetation indices have been slightly modified to cope with the almost total absence of deficiency symptoms.
L358: induce instead if induced à “ induced” has been replaced with “induce”
L359: probably due to… Please add “due” à the suggestions have been followed
Rev comment: “L361: here is a contradiction to what was written above in L337: the Mg threshold is 2 mg g, but Mg concentration in the leaves are far above that threshold. Hence, the term “moderate controlled Mg deficiency” is not accurately used. There was no deficiency in the study. It would be an idea to include optimal Mg ranges for red beet, if available.”
Our response: The terms Mg-deficiency has been replaced with the term “Sub-optimal Mg supply” and the recommended rate of Mg fertilization has been indicated in M&M section.
L373: “…resulted modified…” What is meant here? The N-metabolism was modified? Please rephrase. à now Line 404: the sentence has been rephrased.
Conclusions
L378: “led” instead of “leaded” à now Line 408: “leaded” has been replaced with “led”

Reviewer 2 Report
1. The study was conducted as a pot experiment with a mixture of coconut coir, sand and perlite in greenhouse. Would be the obtained trends analogous under field conditions? It should be discussed more detailed.
2. Several indices related to chlorophyll content were analyzed. A deeper analysis would be needed, which of these indices would be more important. How are they interrelated? Perhaps a correlation can be calculated.
3. Significant differences in tables are marked with uppercase and lowercase letters. Major differences in the tables are capitalized and lowercase. However, this is not explained in the notes. The sentence “Means followed by different letters significantly differ (Fisher’s LSD, p ≤ 0.05)” is repeated twice in the Table 1.
Author Response
Cover letter – response to Reviewer
Manuscript ID: agronomy-551309
Dear Reviewer,
thank all of you very much for the very important and useful suggestions made in order to improve our manuscript.
Follows the response point by point.
We have also highlighted in red all the modifications made along the MS, including Tables and Figures.
We really hope that you will find exhaustive the modifications made, to have the paper published in “Agronomy”
Thank you very much for your cooperation
Sincerely yours,
Michele Pisante
Point by point response to Reviewer 2:
Rev comment: “The study was conducted as a pot experiment with a mixture of coconut coir, sand and perlite in greenhouse. Would be the obtained trends analogous under field conditions? It should be discussed more detailed.”
Our response: The study was conducted under controlled environments (i.e. climate and soil) in order to assess precisely the effect of lack of magnesium. Under field conditions we expect very similar behavior although some additional variable could interfere with the system plant-soil-environment (see for example soil microbiota).
Rev comment: “Several indices related to chlorophyll content were analyzed. A deeper analysis would be needed, which of these indices would be more important. How are they interrelated? Perhaps a correlation can be calculated.”
Our response: As also suggested by Rev1, a part related to VIs has been added in the Introduction section. Moreover, the correlation among VIs has been calculated and reported in the text in the Discussion section (see Lines 389-393).
Rev comment: “Significant differences in tables are marked with uppercase and lowercase letters. Major differences in the tables are capitalized and lowercase. However, this is not explained in the notes. The sentence “Means followed by different letters significantly differ (Fisher’s LSD, p ≤ 0.05)” is repeated twice in the Table 1.”
Our response: The meaning of upper- and lower-case letters have been added in Table captions. The repeated sentence has been deleted from Table 1.
In addition, further modifications made on MS in order to improve English grammar and syntax have been highlighted in red all along the sections. 4 new additional references have been added; 1 old one has been deleted.
